

# A new magnetic observatory in the Arctic - Pituffik Space Base (PIF)

Anna Willer[1], Lars William Pedersen[1], Jan Oechsle[1], Marie Vigger Eldor[1]

[1]DTU Space, Technical University of Denmark, Kgs. Lyngby, 2800, Denmark

*Correspondence to*: Anna Willer (anna@space.dtu.dk)

**Abstract.** Pituffik Space Base is a newly established geomagnetic observatory located in the high Arctic - a region with sparse geomagnetic observational coverage. Positioned within the polar cap at Quasi-Dipole geomagnetic coordinates 83.19° latitude and 25.25° longitude, the observatory provides valuable data from a geophysical significant area. The observatory was carefully designed and constructed using non-magnetic materials with special considerations for the Arctic conditions. All building components were pre-assembled and tested in Denmark before being disassembled, shipped, and reassembled at the remote site in northwest Greenland. This paper presents the observatory design, and an analysis of over one year of operational data, including baseline stability, temperature control, and calibrated vector and scalar magnetic field measurements. The results demonstrate stable baselines and high data quality, confirming the observatory's readiness for data-distribution to the international geomagnetic community.

## 1 Introduction

There are approximately 140 geomagnetic observatories worldwide that contribute to the continuous monitoring of Earth's magnetic field (INTERMAGNET, 2025). In the high northern latitudes, observatories are sparsely distributed, making each station particularly valuable for geomagnetic research and modeling. One of the northernmost observatories is the Qaanaaq (THL) observatory, located in northwest Greenland at a Quasi-Dipole geomagnetic latitude of 84.05°. Since its establishment in 1956, THL has provided high-quality geomagnetic data and plays a crucial role in generating the IAGA-endorsed Polar Cap (PC) index. However, the observatory will likely be magnetically disturbed due to construction activity near the current site in the coming years. To ensure the continued availability of reliable data from this region, a new magnetic observatory has been established at Pituffik Space Base (PIF), situated nearby at Quasi-Dipole geomagnetic coordinates 83.19° latitude and 25.25° longitude.

## 2 Observatory design, construction and installation

The financial funding for the observatory buildings was ensured in 2021, as part of a larger project named GIOS (Greenland Integrated Observing System), supported by the Danish Agency for Higher Education and Science. The buildings were preassembled near the Brorfelde magnetic observatory (IAGA code BFE), in Denmark, allowing for material testing during construction to ensure magnetic cleanliness. Once completed, the structures were disassembled, packed into containers and





shipped to Pituffik Space Base in 2022. Due to the short summer season and rough autumn weather in the region, the assembly of the buildings was postponed to 2023. Figure 1 shows the exterior view of the observatory and the interior of the absolute building to the right. A schematic layout of the observatory, including approximate dimensions and distances, is shown in Figure: 2.

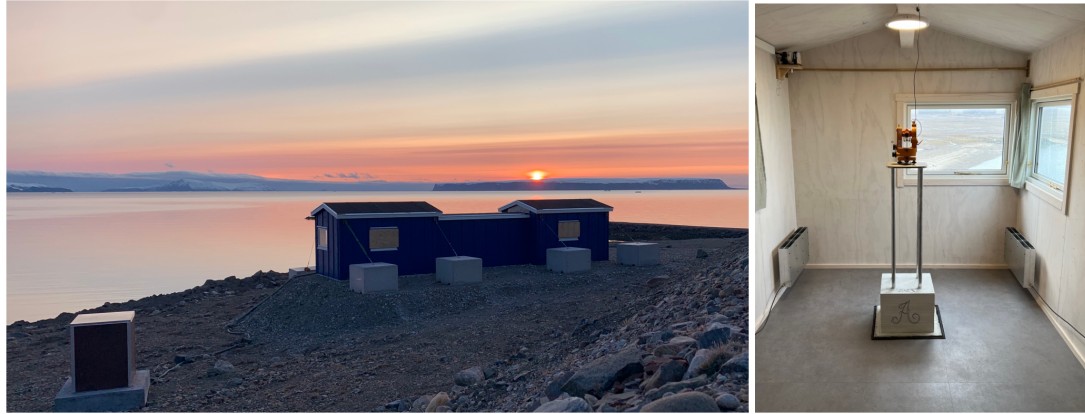

**Figure 1: The Arctic magnetic observatory at Pituffik Space Base, with IAGA code PIF. Left: Exterior view of the observatory. The**
**variometer enclosure is visible in the far-left corner, with the absolute building in the centre connected via a corridor to the electronics building on the right. Right: Interior of the absolute building, showing the fluxgate theodolite mounted on Pillar A (centre) and the scalar instrument located in the upper left corner on Pillar F.**

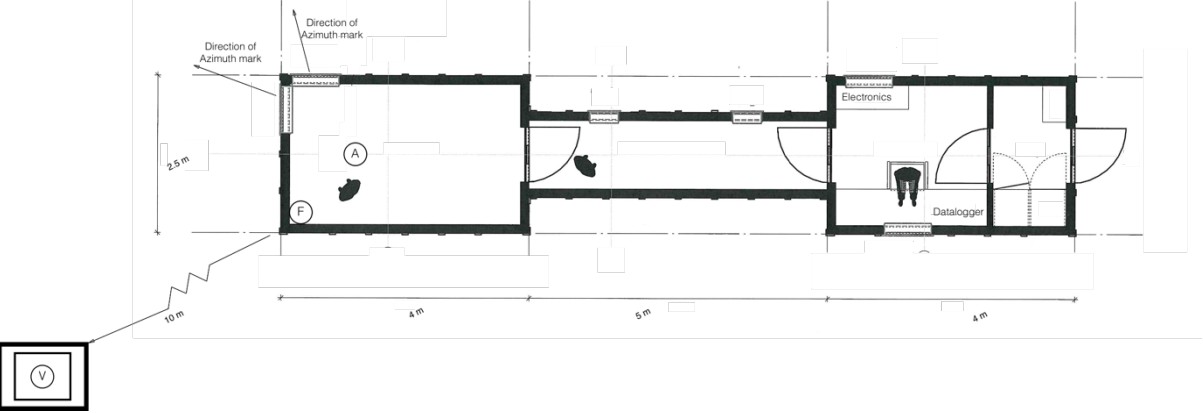

**Figure: 2 Schematic layout of the magnetic observatory at Pituffik Space Base (PIF). The electronics building (right) houses the**
**variometer and scalar electronics and associated data loggers. Magnetic items such as shoes, jackets, and tools are removed and stored in the entrance area. The absolute building (left) contains the scalar magnetometer at Pillar F and the fluxgate theodolite at Pillar A, where absolute magnetic field measurements are conducted. A connecting corridor between the two buildings allows the observer to access the absolute house without exposure to outdoor conditions. The variometer is located approximately 10 meters from the absolute building, installed in an insulated enclosure on Pillar V.**

## 2.1 Reference mark

Absolute measurements with the fluxgate theodolite require precise knowledge of the instrument's orientation in the horizontal plane. This is achieved by referencing horizontal angle readings from the absolute pillar to a fixed azimuth mark with a known



angle relative to true north. The azimuth of the reference mark can be determined through solar observations, using a sun filter mounted on the theodolites monocular. The method used at Pituffik Space Base is described in Newitt et al, 1996. Figure 3 shows an example of the sun observation prior the construction in Pituffik Space Base, above Pillar A, including one of the reference marks. The figure includes an illustration of the angles, including the azimuth to the reference mark.

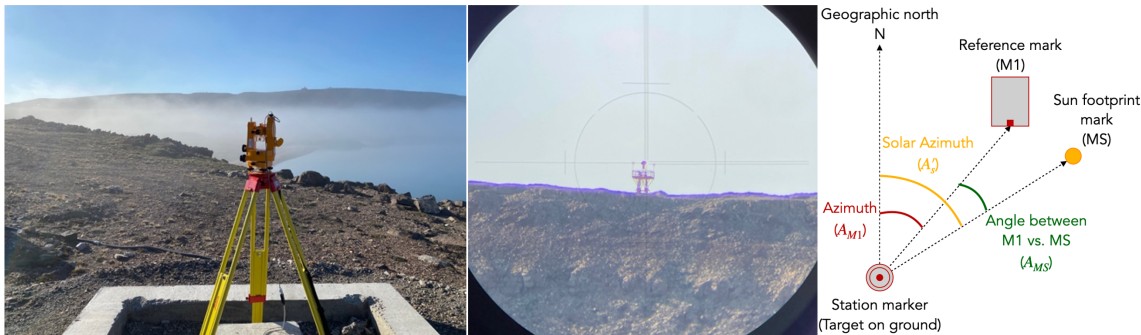

**Figure 3: Sun observations for azimuth determination. Left: The fluxgate theodolite mounted above the Pillar A foundation at PIF, aligned toward a distant reference mark located several kilometres away. Centre: View of the reference mark through the theodolite's monocular. Right: Illustration (by Marie V. Eldor) showing the key angles involved in a sun observation. AM1 is the essential azimuth angle for accurate declination measurements.**

## 2.2 The pillars

The foundations for the variometer pillar (Pillar V) and the declination–inclination pillar (Pillar A) are anchored several meters down to the bedrock and reinforced with fiberglass to ensure long-term stability. Pillar A is constructed with a layer of aerated concrete blocks above the foundation, supporting an aluminium stand with a top plate for mounting the fluxgate theodolite. The scalar instrument (Pillar F) is mounted on a simple shelf located in the upper corner of the absolute house. The scalar difference between Pillars F and A was determined through simultaneous measurements using a GSM-19 scalar magnetometer placed at Pillar A.

## 2.3 Special considerations for Arctic observatories

Due to extreme weather conditions, the presence of polar bears, and the remoteness of the site, special considerations are necessary for Arctic observatories. In Pituffik Space Base, personnel typically carry portable, mobile radio devices to ensure reliable communication and to receive alerts - for instance, in the event of sudden, severe weather. The radio devices are magnetic but do not interfere with scientific measurements when kept in the electronics building. Because the absolute and electronics buildings are connected via an enclosed corridor, observers can access the communication devices and maintain contact without going outdoors. For safety, the observatory's main door is equipped with a polar bear-resistant handle made from non-magnetic materials. The main door, corridor, and both buildings also have windows that allow personnel to visually check for polar bears. Additionally, the windows in both buildings can be opened and serve as emergency exits if necessary.



## 3 Equipment specification

The observatory is equipped with DTU Space three-axis fluxgate magnetometer (model FGM-FGE, suspended version), a GEM Systems Overhauser scalar magnetometer (model GSM-90), and a MinGeo Magrec-4C data logger paired with an ObsDaq 24-bit high-speed analogue-to-digital converter. Additionally, a MinGeo fluxgate theodolite (converted Zeiss THEO 020) equipped with DTU Space DI Magnetometer Model G is used for declination and inclination measurements.

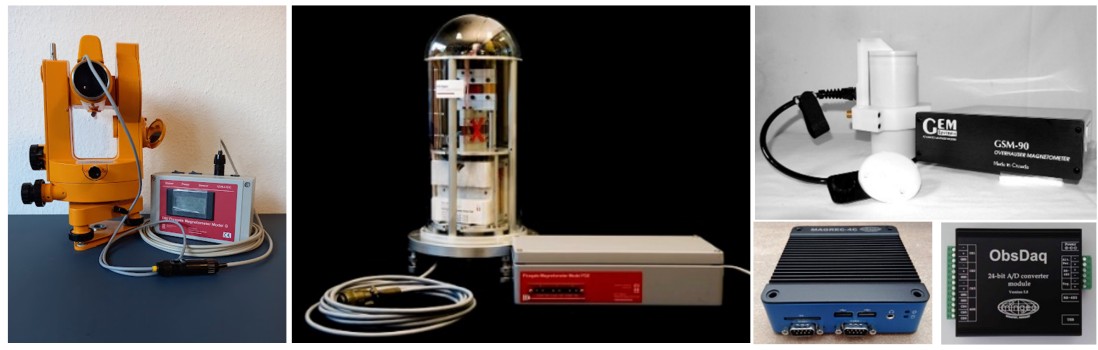

**Figure 4: DI instrument: Fluxgate theodolite (converted Zeiss THEO 020) with DI Magnetometer Model G (to the left, picture from DTU Space), Variometer: 3-axis Fluxgate Magnetometer Model FGM-FGE (middle, picture from DTU Space), Scalar instrument: GEM Systems Overhauser GSM-90 Scalar Magnetometer (to the upper right, the picture is from GEM systems) , Datalogger: MinGeo Magrec-4C with ObsDaq 24-bit fast A/D converter (the pictures are from MinGeo).**

## 4 From variometer data to magnetic observatory data

A variometer station typically consists of a three-axis magnetometer designed for continuous geomagnetic recording, without strict requirements for absolute accuracy. It captures short-term variations with high resolution, while the baselines may drift over time. These baselines are essential to derive the full vector magnetic field at the variometer station and can be determined either through an absolute measurement, i.e., observations of the total field strength, inclination, and declination, or by using geomagnetic model data. The magnetometer at Pituffik Space Base was installed in 2013 and has operated as a variometer

station since. It is mounted on a well-founded pillar, equipped with temperature regulation, and housed within an insulated, non-magnetic protective enclosure. The long-term data series from this variometer has demonstrated good stability over the years, confirming both the instrument and its location as well-suited for observatory magnetic measurements. It requires high-quality data to reach observatory status, which in turn demands quality-controlled data, temperature control and frequent absolute measurements. These absolute measurements are used to solve the baseline equations, and regular absolute

measurements ensure accurate and reliable data (Hansen, 2020). This section presents the absolute measurement results, temperature stability, and a basic validation of the calibrated data.



## 4.1 Absolute data

The absolute measurements were performed in accordance with the procedure described by Lauridsen (1985). All three baselines appear stable over the observed period, as can be seen in Figure 5. The baseline for the horizontal component sensor (H0) remains consistent at approximately 4389 nT ± 2 nT. The declination baseline (D0) varies with approximately 328.7° ± 0.05°. The vertical component baseline (Z0) is stable around 56036 nT ± 1.5 nT throughout the period.

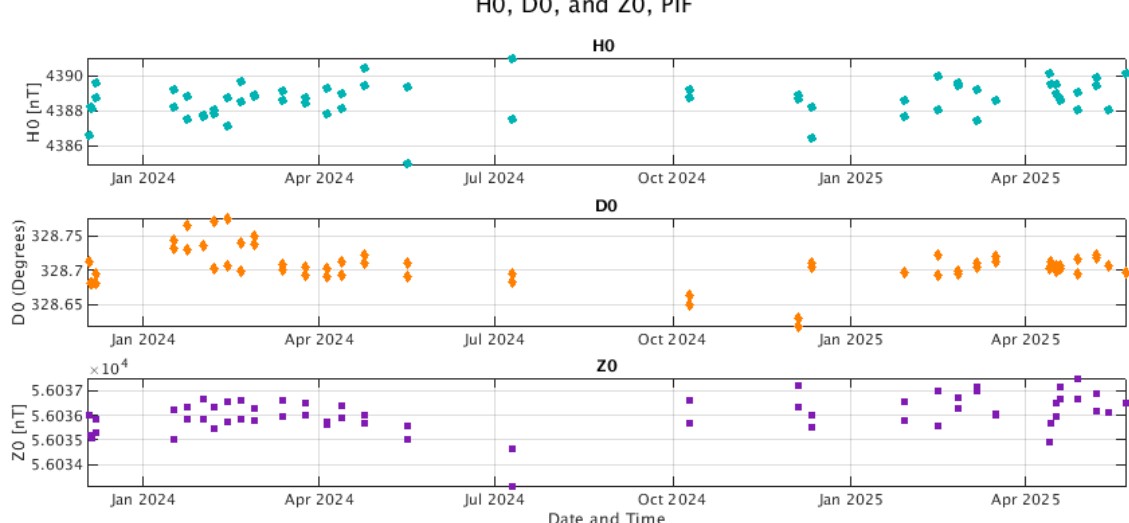

**Figure 5: Baseline values for the magnetic observatory from December 2023 to May 2025. The top panel shows H0, the baseline for the local geomagnetic northward (horizontal) component, in nanotesla (nT). The middle panel shows D0, the baseline for magnetic declination in degrees. The bottom panel shows Z0, the baseline for the vertical component of the magnetic field, also in nanotesla (nT).**

## 4.2 Temperature stability

Temperature is continuously monitored and regulated for both the magnetometer sensor located at Pillar V and the electronics housed in the electronics building. Both systems maintain stable temperature conditions throughout the year. Figure 6 presents a representative example, showing temperature data for February 2024. During this period, the sensor temperature typically fluctuates around 16.5°C ± 0.5°C, while the electronics temperature remains near 31°C ± 1.5°C.



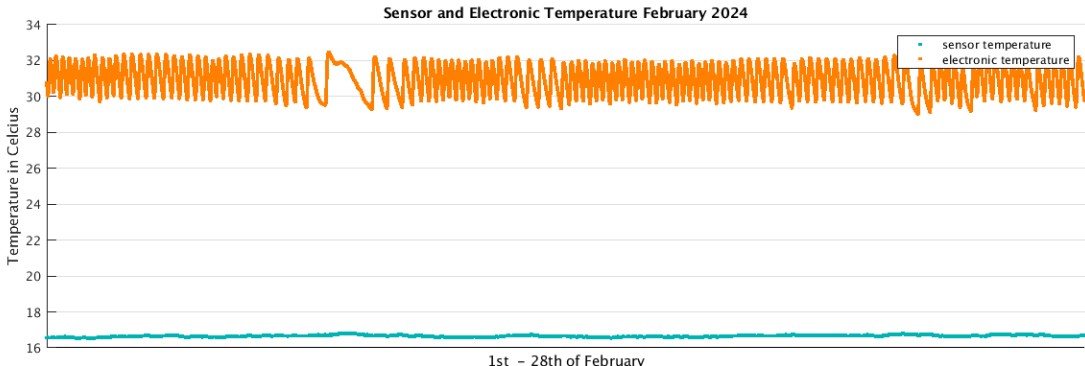

**Figure 6: Sensor and electronic temperature data (1 sec resolution) for the 3-axis FGM-FGE magnetometer at PIF in February 2024.**


### 4.3 Validation of an example day

The full vector field has been calibrated based on the baseline measurements in Figure 5. As an example, a single day of data from the 1st of October 2024 is shown in Figure 7. According to the CHAOS-8.1 model (Kloss et al, 2024), a declination around -37 degrees is expected nearby PIF in 2025 (see the map in Figure 9 with the declination in red), fitting well with the

declination plot (D plot, Figure 7) that varies around -36 degrees. The model also predicts a scalar field around 56300 nT, again similar with what the measurement at the observatory gives, see an example in Figure 8, showing values around 56400 nT. Exact agreements between model and measurements are not expected, due to the local magnetic field in Greenland. Figure 8 shows an example of the scalar measurements (Fs) and calculated scalar field based on variometer data (Fv). The two datasets, Fs and Fv, show generally good agreement. However, small discrepancies are observed during periods of geomagnetic

activity. The example from October 1st 2024 shows differences within ±0.5nT, and the difference can be even larger during rapid field variations. These differences may be attributed to limited time synchronization accuracy between the instruments or to localized induced magnetic fields that vary between the two measurement sites. Further investigation is required to determine the exact cause and to identify potential improvements in measurement accuracy and system performance.



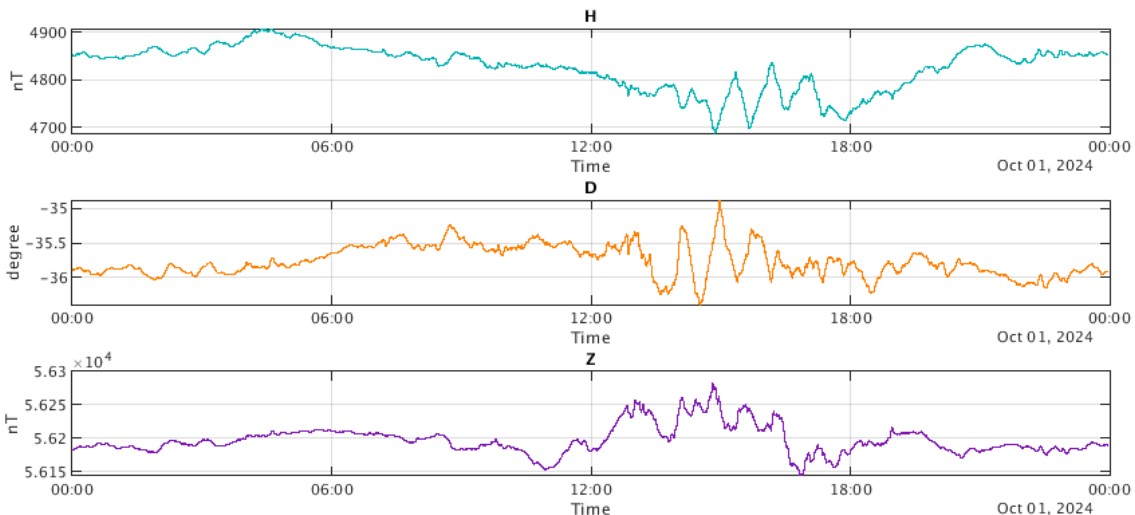

**Figure 7: One day data with 1-sec resolution, October 1, 2024. Top: The horizontal intensity, H, Middle: The declination, D. Bottom: The vertical component Z.**

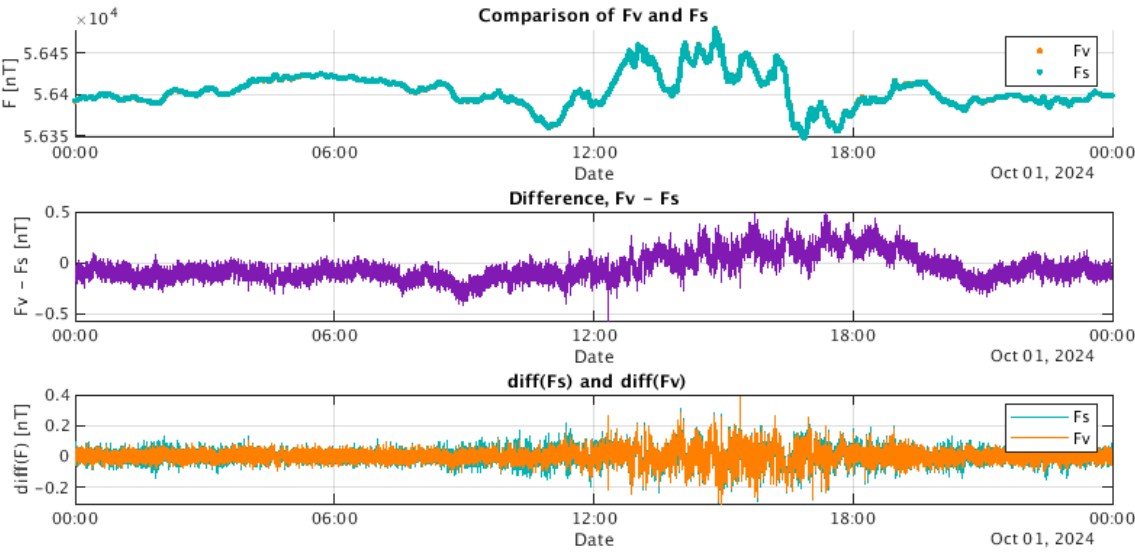

**Figure 8: Comparison of 1-sec scalar magnetic field strength over one day in October 2024. Top: Measured scalar values (Fs), including the pillar offset correction, are shown alongside the calculated scalar field derived from variometer data (Fv). Middle: Difference between the two datasets (Fv-Fs). Bottom: Differences between adjacent scalar measurements, illustrating short-term variability.**



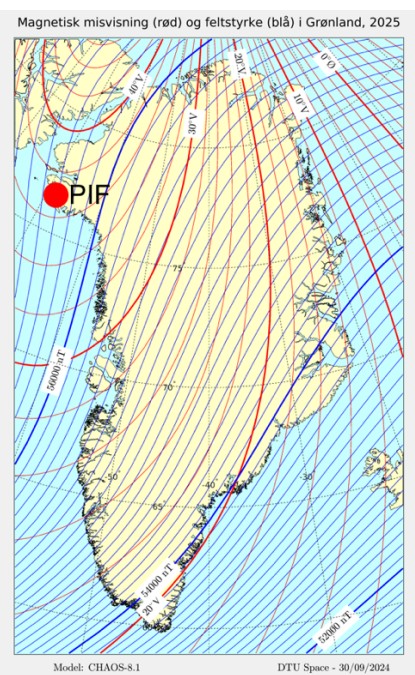

**Figure 9: a map with the declination (red) and total field strength (blue) over Greenland, generated using the CHAOS-8.1 magnetic field model. This model was developed by Clemens Kloss, Chris Finlay and Nils Olsen from DTU Space, who also provided the map.**
**Additionally, the location of Pituffik Space Base is marked on the map with a red circle.**

## 5 Conclusion

The new Arctic magnetic observatory at Pituffik Space Base in Greenland is strategically located in a region with sparse coverage of geomagnetic observatories, contributing valuable data from a critical high-latitude area. The site was carefully selected to ensure an undisturbed environment with minimal magnetic interference. The observatory is thoughtfully designed
and well-constructed using non-magnetic materials, and it is equipped with high-quality instruments and data acquisition hardware, including temperature-controlled magnetometers.

Routine absolute measurements indicate stable baselines throughout the observation period. The comparison between scalar magnetometer readings and scalar values calculated from variometer data shows overall good agreement. Minor discrepancies during periods of magnetic activity may be attributed to induced magnetic fields or timing inaccuracies between instruments;
these effects warrant further investigation to improve data consistency.

Overall, the observatory delivers high-quality magnetic field data and is well-prepared for integration into the global geomagnetic observation network.




## 6 Author contribution

LWP, JO and AW designed the observatory, tested all material during construction, worked on the interior (for example Pillar A), installed the scalar magnetometer and the fluxgate theodolite, made the first absolute measurements, and analysed the data. MVE contributed with text and illustration to section 2.1 Reference mark. AW drafted the manuscript, with all co-authors
contributing to its editing.

## 7 Acknowledgement

The authors wish to acknowledge the Danish Agency for Higher Education and Science, that financially supports the GIOS (Greenland Integrated Observing System) project.

The authors used the AI tool ChatGPT (https://chat.openai.com) to assist with improving the English language in specific parts of the manuscript.

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
