# Peer review of "Pituffik (PIF), a new magnetic observatory in the Arctic"

_EGUsphere, 2025_

## Author Response (AR1)

*Dear all,*

*We would like to express our sincere gratitude to the referees for their valuable input, insightful suggestions, and constructive corrections. We also thank the handling associate editor for providing guidance throughout the review process.*

*The article has been revised in accordance with the referees recommendations. Detailed responses are provided below inline with the referees comments. Each response is presented in italics and begins with the prefix 'Answer:'.*

*Additionally, we have changed the naming of the observatory from Pituffik Space Base to Pituffik, to be consistent with the in-house geomagnetic observatory documentation and metadata.*

*The updated article with track-changes has been uploaded, along with a clean version.*

*Kind regards,*

*Anna Willer, Lars W. Pedersen, Jan Oechsle, and Marie V. Eldor*

**Referee 1**

The manuscript "A new magnetic observatory in the Arctic - Pituffik Space Base (PIF)" by Anna Willer, Lars William Pedersen, Jan Oechsle1 and Marie Vigger Eldor is a very nicely written account of a new geomagnetic observatory in the Arctic and contains valuable information for the geomagnetism community. I have only few comments.

Figure 2

Improve resolution of the text (e.g. „Direction on Azimuth mark", „Electronics"). The same applies to Figure 5, Figure 6, Figure 7, Figure 8, Figure 9.

*Answer: We have improved the text resolution on all figures.*

Line 49

What exactly is meant with the method described in Newitt et al.? Is it the method described on page 36 of Newitt et al?

*Answer: Yes, we used the method described in chapter 5.4.2 Determination of azimuth from sun observations (page 34 - 38), this information is added in the reference.*

Figure 3

Is the central photograph showing M1?

What is a "Sun footprint mark"?

*Answer: Yes, the central photograph show M1, this is now included in the caption. Sun footprint mark is not a good description. Thank you for this question, we have replaced it with "Sun position".*

Is there a reference to Figure 4 in the text?

*Answer: A reference to Figure 4 is added in Equipment specification.*

Line 118

Kloss et al, 2024 -> Kloss et al., 2024

*Answer: This is now corrected*

Figure 9 is mentioned in the text before Figure 8, could you move the sentence referring to Figure 9 further down?

*Answer: We have combined Figure 8 and 9 into one figure, to improve the comparison between the model and an example day of data.*

Line 127

This is not only limited time synchronization accuracy, also the different filtering could play a role: The GSM measures for a few 100 ms at some point after polarization, while the FGE is likely sampled continuously at 100 Hz or so and then properly filtered.

*Answer: Thank you for this information, we have included that the different in filtering might be part of the differences between Fv and Fs.*

**Referee 2**

The article describes the newly established Pituffik Space Observatory. Data from this observatory, obtained over a year of observations, are analyzed. The change in baselines, temperatures, and the difference between measured and calculated values of the magnetic field induction vector are shown.

Since the observatory is located at high latitudes, its data are very valuable for research in this region. This is due to the fact that the number of observatories at high latitudes is limited.

Remarks.

1. The authors do not state in the article how often absolute measurements are made. Figure 5 shows that only 5 observations were made from May to December 2024. This is too few to claim that the baselines are stable.

   *Answer: The aim is to have weekly measurements, according to INTERMAGNET standards. That is not fulfilled for the whole period presented in the article, and we*

*agree that we cannot know if the baselines are stable within the gaps. We have rephrased the sentence to address this.*

2. In order to better see how the baseline values change, I would advise the authors to plot the adopted values next to the measured values.

   *Answer: It is a good idea, we have now implemented that.*

3. Figure 6 shows how the temperature of the magnetometer placed on the V-pole and the electronics placed in the building changes. The stability is very good. However, this is only shown for February 2024.
4. Is the same stability observed throughout the year?

It would be good to provide a graph of the magnetometer temperature for the entire period.

   *Answer: We have included a plot for a whole year, to show the temperature variation.*

1. Figure 8 above only shows the Fs graph and not the Fv graph.

   *Answer: It actually shows both, but the values are almost identical, so the Fv graph is almost impossible to see. We have increased the size of the marks for Fv so make it visible.*

2. The article does not mention anything about the displacement of the pole, which is described in the caption to Figure 8.

   *Answer: The pillar differences is briefly mentioned in the end of chapter "2.2 The Pillars"*

3. Figure 8 shows the difference between Fs and Fv for October 1, 2024, which is quite good. However, this difference is not shown for the entire period. The article only mentions that during geomagnetically active periods the difference between Fs and Fv is larger. And by how much?
4. It would be good to provide a graph of the difference between Fs and Fv for the entire observation period.

   *Answer to 3. And 4.: We tried different ways of presenting this, and went with a 5-days presentation from October 10th to 15th 2024. This period includes an intense geomagnetic storm, capturing high differences between the Fv and Fs during activity, and also shows the general difference close to zero during days with less activity.*

5. The caption to Figure 1 indicates that a scalar instrument is placed on pillar F. Figure 1 shows that F is not a pillar, but a shelf and only the GSM-90 sensor is placed on this shelf.

   *Answer: Yes, the wording "pillar F" is misleading, we have changed the description to location F.*

6. The article does not provide a graph of the difference between the geomagnetic field induction vector on pillar A and shelf F.

7. How often is the difference between the geomagnetic field induction vector on pillar A and shelf F checked?

*Answer to 6. And 7.: We measured the scalar field difference between pillar A and location F using a GSM19 on pillar A – but we have no plans of measure it more frequently than yearly (when DTU staff visit the observatory).*

Advice for employees of the PIF geomagnetic observatory.

GSM-90 is placed on shelf F, which is connected to the observatory building and is not as stable as pillars A and V. Therefore, the increase in the difference between Fs and Fv may also be associated with meteorological conditions. I would advise during absolute measurements to measure the geomagnetic field induction vector on pole A using GSM-90. With this method of conducting absolute measurements, the difference between Fs and Fv will be much smaller.

*Answer: We are thankful for your advice and might consider implementing this routine when our observers are more familiar with the instruments.*